# Relationship between Panic Buying and Per Capita Income during COVID-19

**Hugo T. Y. Yoshizaki [1,2]**, **Irineu de Brito Junior [1,3,\*]**, **Celso Mitsuo Hino [2]**, **Larrisa Limongi Aguiar [1]** and **Maria Clara Rodrigues Pinheiro [2]**

1   Graduate Program in Logistics Systems Engineering, São Paulo University, São Paulo 05508-010, Brazil; hugo@usp.br (H.T.Y.Y.); larissalaguiar@usp.br (L.L.A.)

2   Department of Production Engineering, São Paulo University, São Paulo 05508-010, Brazil; cmhino@usp.br (C.M.H.); mariaclarapinheiro@usp.br (M.C.R.P.)

3   Environmental Engineering Department, São Paulo State University, São José dos Campos 12247-004, Brazil

\*   Correspondence: irineu.brito@unesp.br; Tel.: +55-12-99702-9119

**Abstract:** Panic buying and hoarding express common human behavior in times of crisis. Early in COVID-19, as the pandemic crisis intensified, toilet paper was one of the emblematic cases of panic buying. Using a Geographic Information System (GIS) to cross official per capita income data and real toilet paper transactions obtained from groceries spread around the city of São Paulo (Brazil), this study compares sales levels during the period in which panic purchases took place to the sales levels off that period. As expected, that data disclose noticeable panic buying. Regression analysis reveals that there is a significant positive correlation between average income per capita and panic buying. The results also indicate that panic buying happens in every income class, including low-income ones and contribute to enhancing the understanding of demand behavior during periods of crisis.

**Keywords:** panic buying; hoarding; per capita income; vulnerability; COVID-19; São Paulo; grocery

## 1. Introduction

In 2020, humanity has faced a global health crisis called COVID-19 caused by a virus: the severe acute respiratory syndrome coronavirus 2 (SARS-CoV-2). The pandemic is not just a medical issue; it has other implications in the daily life and livelihoods of many people [1] and also in socioeconomic and psychosocial aspects. In the pandemic response, population behavior is a key subject [2].

A pandemic is a natural disaster of the biological subgroup caused by exposure to living organisms according to the EM-DAT (The International Disaster Database) classification [3]. During disasters and healthcare crises, people under uncertainty about the future undertake behavioral changes [4]. These changes could be negative manifestations of violence, poor investment decision making, herd mentality, purchasing habits, and panic buying [4,5].

Panic buying is an instinctive reaction [6] and a common human response, which is not caused by shortages, but rather by a fear of simply running out of supplies. When the population is aware of shortages, they start to buy out of panic and hoard to mitigate the risk of future scarcities [7], which may cause shortages [8]. Hoarding is a form of behavior in which consumers cope with scarcity [9], accumulating safety stocks and buying more than their needs [10].

In March 2020, as the pandemic crisis intensified and quarantines and lockdowns were imposed, panic spread mainly by social media [11], causing people to flock to retailers to buy basic products, particularly toilet paper. In some countries, this behavior resulted in severe stockouts of toilet paper [12,13] and limits to purchase were imposed [14]. Struggles and hysteria also happened in supermarkets and groceries [15].

Regarding humanitarian and aid operations, this behavior is addressed, for example, in the Sphere Handbook, which recommends that the issue should be managed and the traders be prepared for it [16]. Aid organizations, such as food banks, may run out of products because people stop donating. Vulnerable populations lose their purchasing power [8] due to inflation [4].

In São Paulo city (Brazil), quarantine was decreed on 22 March. A few days earlier, the disclosure of the first cases of the disease in Brazil, as well as news in the media and social networks showing excess purchases and hoarding in other countries, caused a rush to supermarkets in the city to buy toilet paper [17].

São Paulo is a megacity of 12 million inhabitants. As in many large cities from developing economies, its different geographic districts present large discrepancies in social indicators, as educational levels, service availability, types of housing, occupancy rate, and income distribution. Households located in more central regions have better social indicators than those located in peripheral regions [18]. However, the effects of panic buying and hoarding spread throughout the city, no matter those differences.

This paper aims to compare and to evaluate the magnitude of these effects in distinct regions of the city, analyzing whether the level of panic buying behaves differently among regions with different average income levels. Thus, the research question is: Is there significant positive correlation between per capita income and panic buying?

The product of choice is toilet paper, as there was plenty of coverage by both mass and social media, demand has effectively peaked during that period, it is a classic example of small demand elasticity [19], and there were no supply shortages on groceries, allowing the effective comparison of sales volume during and off the panic buying period.

The authors correlated actual transaction data from 144 grocery outlets spatially distributed over the city of São Paulo to their average per capita income from their immediate influence area or zone (immediate surroundings of those stores). A Geographic Information System (GIS) set up each influence zone using a buffer tool to calculate the average income from detailed household information from the Brazilian 2010 Census (Brazilian Institute of Geography and Statistics—Portuguese acronym IBGE) [20]. Panic buying was measured at the store level by calculating the ratio between average sales during the ten-day panic buying period to the overall sample period of three months. Next, a regression analysis estimated the correlation between the average income per capita and panic buying.

As far as the authors know, there is no equivalent study in the academic literature, and it is our main contribution to the state of the art. This article also contributes to support retailers and government decisions, as its lessons can be applied to other panic buying items. Retailers can better understand customer behavior in times of crisis and the government, by knowing that even lower-income populations practice hoarding, could act to prevent supply chain disruptions to the most vulnerable residents in pandemics.

The remainder of this text is organized as follows: Section 2 presents a brief literature review; Section 3 lists methods and materials. Section 4 offers paper results. Discussions and concluding remarks are in Section 5.

## 2. Brief Literature Review

### 2.1. Consumer Behavior in Extreme Events

To support the study, works related to the panic buying phenomenon have been searched on online academic databases Scopus, Web of Science, and Google Scholar. Aiming to identify a theoretical frame for this research, combinations of the words "panic buying", "hoarding", and "stockpiling" were used to collect articles, congress works, and reports, and the authors filtered the material aligned to the scope of this work. Most of the findings were in the psychology and medicine fields examining and discussing its origins and causes [21–23]. Therefore, there is a lack of literature about consumer behavior changes

caused by extreme events and their impacts on the supply chain entities, and literature building a relation between panic buying and per capita income is also scarce but growing.

There is also a rapidly emerging academic literature related to the pandemic's effects [24], including topics encompassing all aspects of human affairs, such as shopping, household violence, food consumption, circular economy, socio-economic implications, energy consumption, mobility, and so on [1,2,4–8,11,12,21–23,25–27]. This is a vast and fascinating subject *per se* and it is out of the main focus of this paper. There are some previous works to the COVID-19 pandemic that report a shortage of goods caused by panic during disasters, such as the lack of gas in Florida during the Irma hurricane event [28], in which the demand increased by more than 105%. Other authors studied the fuel purchase in the UK during two rupture periods using agent-based simulation [29]. The same technique has been employed to analyze panic diffusion over social media [30]. In Japan, the theme was explored through monitoring consumer purchase behavior [31]. That study applied empirical analysis to identify the consumption pattern for several products to point out those with panic buying evidence. Additionally, it examined the characteristics of families engaged in panic buying using probit regression.

The sequence of earthquake events in 2010 and 2011 in New Zealand was examined using a qualitative exploratory study to understand the consumer's experience related to retail shopping [32]. Then, a discussion of how people responded to that natural disaster was conducted, showing evidence of stockpiling after the quakes, and some participants suggested coping strategies to endure the situation. Additionally, China experienced salt panic buying in 2011, resulting from the Japanese nuclear leak triggered by an earthquake. [33], with a special insight on social network analysis theory, focused on information dissemination.

As a nation with a propensity for earthquakes, Japan encourages its households to stock foods for the immediate post-disaster period. For instance, the Great Hanshin (Kobe) earthquake revealed that, despite the existence of public stock and an effort from retailers donating food, household stocks are the major source of supplies [34]. The reason is that distributing goods requires time and a transportation setup. Hence, stockpiling is also considered a preparedness strategy. That paper analyzed stockpiling by applying a questionnaire and collecting data about socio-economic characteristics, too. Then, the study showed that 48% stockpile food and water, while 35.1% stockpile only food. A probit analysis found that high-income households are 6.5% more likely to stockpile.

Thus, the uncertain conditions faced by consumers are pointed out as one of the propellers to the tendency to stockpile huge quantities of items. This is a strategy to mitigate the risk of future failures, frequently observed in supply chains with rupture risk [35]. Hoarding and stockpiling also express a desire to maintain a routine in contrast to the uncertainty that surrounds the epidemics. Those coping strategies cited are an auto preservation deed, and social media has contributed to the propagation of these psychological responses [36].

Stockpiling can be considered speculative accumulation behavior caused by panic [37] where non-coercive social influence has been the most significant aspect; as in [35], customers' decisions are affected by their peers. Social learning impacts purchase decisions in supply chains under risk [35]. Hence, customers evaluate future beliefs by observing other individuals. There is also a recent literature review about panic buying [38] that classifies its cause factors as perception (subdivided into perceived threat and perceived scarcity), fear of the unknown, coping behavior, and social psychological factors (subdivided into social influence and social trust).

## 2.2. Recently Panic Buying Works (During and Before COVID-19 Pandemic)

As said previously, the novel coronavirus quickly spread over the world and caused changes in the consumers' patterns in several countries. The situation has impacted supply chains and an increase in research about that subject. Materials that have been published reporting the panic buying behavior as a consequence of the pandemic explored the correlation between panic buying and behavioral and

social factors [39,40], describing the events in their countries [41]. Other authors tried to evaluate future tendencies [42,43] and also provided discussions on how to prevent or to mitigate panic buying [44].

Generally, the COVID-19 pandemic triggered changes in product consumption behavior in several countries. One specific work [1] studied the correlation between variance in the client's food preferences (purchasing methods, time windows, minimum order requirements, and fees) and trends in the COVID-19 pandemic, that is, growth, decrease, and stability scenarios. Another study [7] analyzed Qatar's food consumption behavior during the pandemic and observed a change in the direction of healthier diets, seeking local foods, as well as a rise in online purchases. Note that the pandemic is influencing purchasing behavior in several aspects; however, in this study, the focus is on materials associated with panic buying and stockpiling.

There are recent reports that have investigated the incidence of panic buying during the COVID-19 pandemic using statistical techniques such as regression models [45], multivariate statistics [46], logistic regression [47], econometric models [12], and big data [48]. Those studies aim to identify the influence of factors on purchasing behaviors, such as fear of a complete lockdown, peer buying, scarcity of essential products on shelves, limited supply of essential goods, and panic buying in the previous period [46], perceived stress, media information and changes in product supply [47] and even moments of rationing policies and intervention policies [48]. Others seek correlation with the level of social distance and social characteristics, such as the number of individuals in a household [45].

It is important to note the influence of media on the spread of panic buying and stockpiling behaviors during the COVID-19 pandemic. Some studies have explored the influence of social media both on triggering panic buying, through content about proof of product unavailability, authorities' communications, and expert opinions [49], as well as on the development of stockpiling behavior, motivated by information on risk perception, institutional communication, and evidence of global uncertainty [50]. Studies were also carried out on media news and its contents, such as potential causes of panic buying, related government actions, and expert opinion [44] and on research data in Google Trends, to establish correlations between increases in COVID-19 cases and increases in online research on panic buying [51].

The occurrence of panic buying triggers the scarcity of goods and affects vulnerable people that have no economic power to stock. Generally, low-income populations are more vulnerable to disaster situations and are less prepared. A former study [52] evaluated the general preparedness of families in Florida facing disasters and concluded that the majority of families were prepared to self-maintenance/subsist in the post-disaster, but higher-income families were better prepared than lower-income families. In the UK, a recent work [53] analyzed panic buying effects in the supply reduction of some products and food security of vulnerable populations during the pandemic period. Although there are studies that discuss the potential impacts of panic buying and disasters on vulnerable populations, there are currently no studies that explore the correlation between panic buying and socio-economic criteria, such as per capita income.

Some news in the media, based on the experiences and observations of grocery store owners and managers in New York (USA) [54] and Cairo (Egypt) [55], also showed that the effect of panic buying occurs differently in neighborhoods according to the local income [54,55]. Nothing was found in the academic literature linking panic buying and income by utilizing quantitative analysis and with statistically significant results. This is the main contribution of this paper.

## 3. Materials and Methods

The research methodology is detailed here and includes four parts: data collection and validation, calculation of per capita income in the influence area for individual stores, and statistical analysis.

Data are from two sources (retailers and Brazilian IBGE) and comprise actual sales data and per capita income. Sales data came directly from three different top retailers, and the authors express their appreciation for sharing such valuable information. One of those retailers has basically small shops (convenience or proximity, located mostly in fuel stations and office/working districts) and

unfortunately, their data were discarded due to chronic stockouts during the ten-day panic buying period; additionally, the very limited in-store availability (only a few units on the shelf) means that this outlet format does not fit well for hoarding. The two other grocery chains, which have stores in the traditional supermarket and supercenter format (700 to 5000 square meters), contributed with a set of 144 locations all over the city and did not have issues with product availability throughout that sample period (24 February to 24 May). As mentioned, full availability (no stockouts) is one of the main reasons for choosing toilet paper as the focus product, so that adequate measurements of hoarding levels could be made. Sales per store are measured in the total number of units sold (encompassing all product types, from low- to high-end brands), since it is not influenced by socioeconomic level.

Per capita household income comes from the official 2010 census data [20] at its lowest level (census sector, a few blocks size). To associate the neighborhood per capita income to each store, the concept of store influence area [56,57] is used: the store influence area contains at least 60% of its customer base (households) and could be roughly approximated by a circle centered in that store. Both retailers still employ a radius of 1 km for supermarkets and 2 km for supercenters for store location analysis, following the Brazilian literature for São Paulo city [56]; this paper uses those radii as a standard. To check for consistency, a sensitivity analysis was performed in the study, varying the influence radius in steps of 0.5 km (see Table 1). To guarantee accuracy, all store coordinates (lat/long) have been manually double-checked, crossing addresses, borough, and Google StreetView. For each store, a buffer zone of influence was drawn in a GIS and the average per capita income determined from the IBGE data base.

**Table 1.** Store influence area radii (buffer size for the geographic information system).

| Store Format | Influence Area Radius (GIS Buffer Size) | | |
|---|---|---|---|
| | Small | Standard | Large |
| Supermarket | 0.5 km | 1.0 km | 1.5 km |
| Supercenter | 1.5 km | 2.0 km | 2.5 km |

Figure 1 shows the spatial distribution (heat map) of the average monthly per capita income by individual census sector (Figure 1a) and the set of influence areas (buffer zones) of all 144 stores (in red) superimposed on the São Paulo Metropolitan Region urban sprawl (Figure 1b), with São Paulo city boundaries in yellow. Picture 1b utilizes standard buffer sizes (see Table 1). From Figure 1, the area covered by store buffers spreads over most of the populated areas of São Paulo and also over a broad range of colors (per capita income) in the heat map.

The authors performed regression analysis to evaluate the correlation between panic buying and the average per capita income of its buffer zone for each store. Statistical analysis was performed in R and Excel and geographic data processing in Maptitude GIS software.

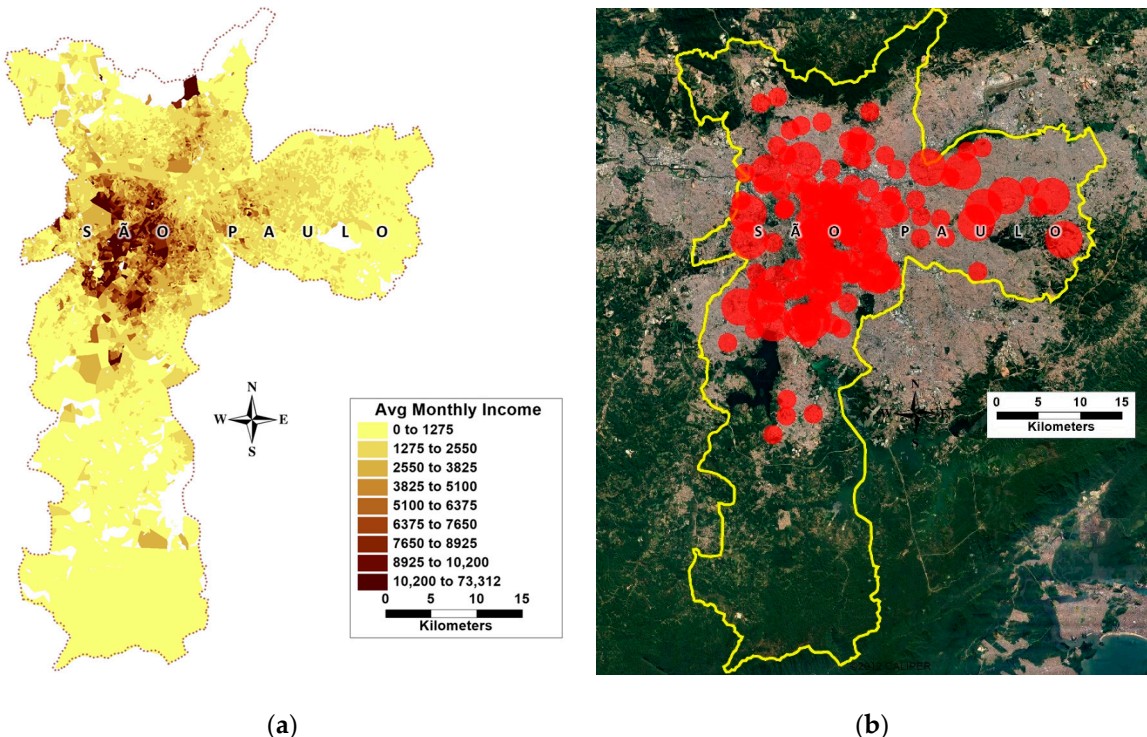

<div align="center">(<b>a</b>)                                                          (<b>b</b>)</div>

**Figure 1.** Panel (**a**) shows the spatial distribution (heat map) of the average monthly income by census sector for the City of São Paulo. Panel (**b**) shows the urban sprawl of São Paulo City (inside the yellow border), with standard size buffer zones superimposed. From both images, the buffer zones cover a significant proportion of the city population and the full range of incomes. (Map data: Caliper® Maptitude Software and Google Earth®, Mountain View, CA, USA).

## 4. Results

### 4.1. Did Panic Buying Happen in São Paulo City During COVID-19?

Media reported panic buying in the city of São Paulo starting on Wednesday, March 11 [17], when the World Health Organization declared the COVID-19 pandemic. Things ramped up on Thursday the 12th, even before the city government imposed the closure of all non-essential services on the 20th and the state government announced the quarantine on 21 March to start on the 24th. People hurried to buy groceries before the quarantine started.

Figure 2 shows the total sales for toilet paper in the city of São Paulo for those two retailers from 24 February to 24 May. This period includes data from before and after the panic buying to allow an overview of the phenomenon. Carefully evaluating all the sales profiles from all the stores in this convenience sample, the authors considered the panic buying period as the ten days from 12 to 21 March. This period is consistent with media reports and with the chronology of events.

In Figure 2, the peak sales on 29 February and 7 March are due to toilet paper promotions that happened every Saturday before quarantine started on the 24th (more than half of the stores traditionally did that to attract people at weekends). As the product is non-perishable and prone to hoarding, this means that panic buying happened even when some customers were already well supplied at home. One important characteristic of panic buying is that sales sustain peak levels if stores have enough inventory and, therefore, stockouts do not occur.

In fact, only three stores (from 144) presented one or two days of stockouts in those ten days; as they were immediately replenished, they were kept in the sample.

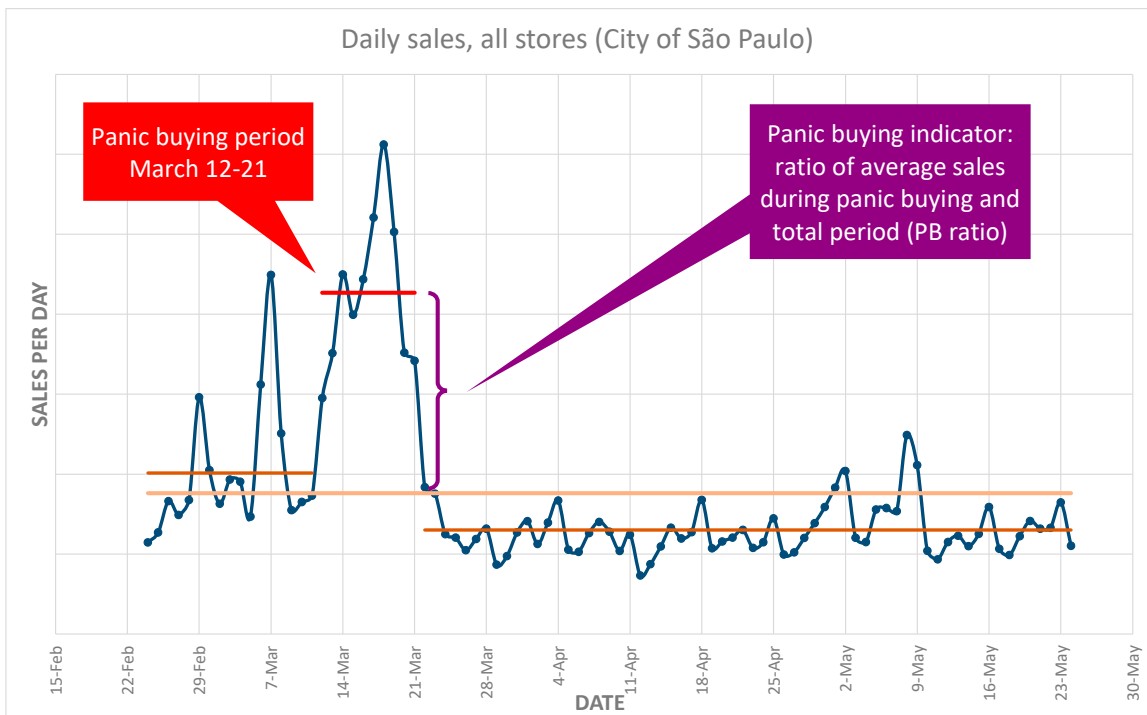

**Figure 2.** Total daily sales (toilet paper units) from both retailers in the city of São Paulo (144 stores). One can observe that sales during the panic buying period are higher than in other periods. Peak sales on February 29 and March 7 are due to toilet paper sales every Saturday before quarantine started on the 24th. Sales figures are not shown to preserve confidentiality.

After checking all store profiles, one clear, quantitative panic buying indicator could be defined: the ratio between average sales during the panic buying period (the red line from 12–21 March) and the average sales in the complete period (the orange line from 24 February to 24 May), the (panic buying) PB ratio (Figure 2). The red lines are the average sales for each segment of the total period (before, during, and after); sales after the quarantine are clearly below pre-COVID-19 levels.

Panic buying happened in São Paulo city.

### 4.2. Does the Sample Adequately Cover the City?

From Figure 1, it is visually apparent that the store buffers cover most of the city and income values. Those 144 stores are scattered all over the urban sprawl, and using different influence buffer sizes, they indeed cover from 34.4% to 67.6% of the entire city population (Table 2).

**Table 2.** Population coverage for different buffer sizes.

| Influence Buffer Radius | Area | Covered Population | Population Coverage |
|:---:|:---:|:---:|:---:|
| 2.0 km | 654 km$^2$ | 7,609,242 | 67.6% |
| 1.5 km | 511 km$^2$ | 6,027,360 | 53.6% |
| 1.0 km | 314 km$^2$ | 3,874,343 | 34.4% |

Obs. São Paulo population is 11,252,204 (Census 2010); total area is 1524 km$^2$.

Calculating the average per capita income for each store buffer (standard radii), the researchers obtained the following distribution (Figure 3). Per capita income (in BRL—Brazilian Real) varies from BRL 770/month to BRL 10,989/month.

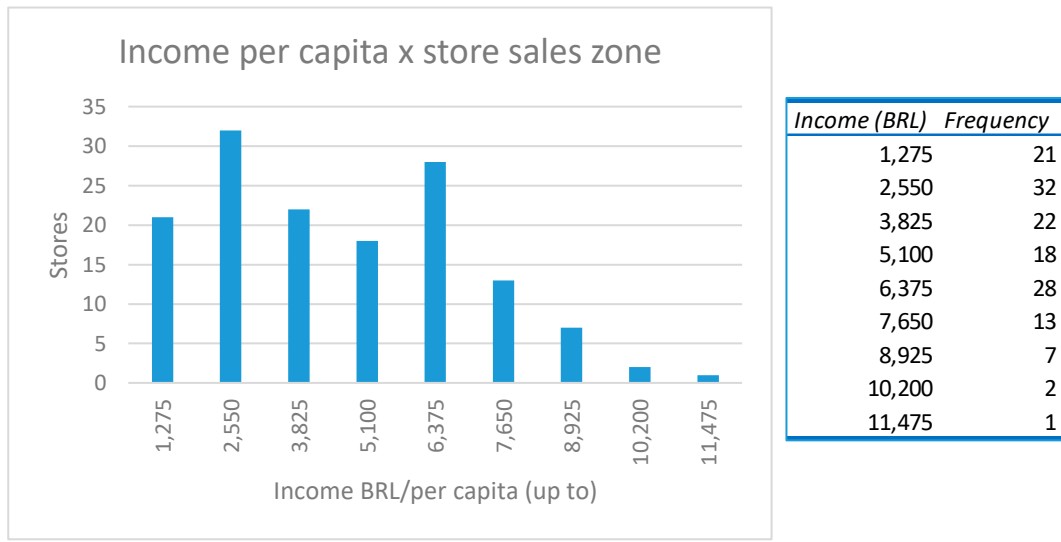

**Figure 3.** Histogram of per capita income for store influence buffer area; all socioeconomic categories are represented. Buffer radii are standard (1 km for supermarkets, 2 km for supercenters).

This convenience sample of 144 stores covers the main city characteristics for this study.

*4.3. Is There a Positive Correlation Between Income and Panic Buying?*

Figure 4 allows nothing that there is a correlation between income and panic buying for all sizes of store influence areas (according to Table 1), with the same distribution pattern.

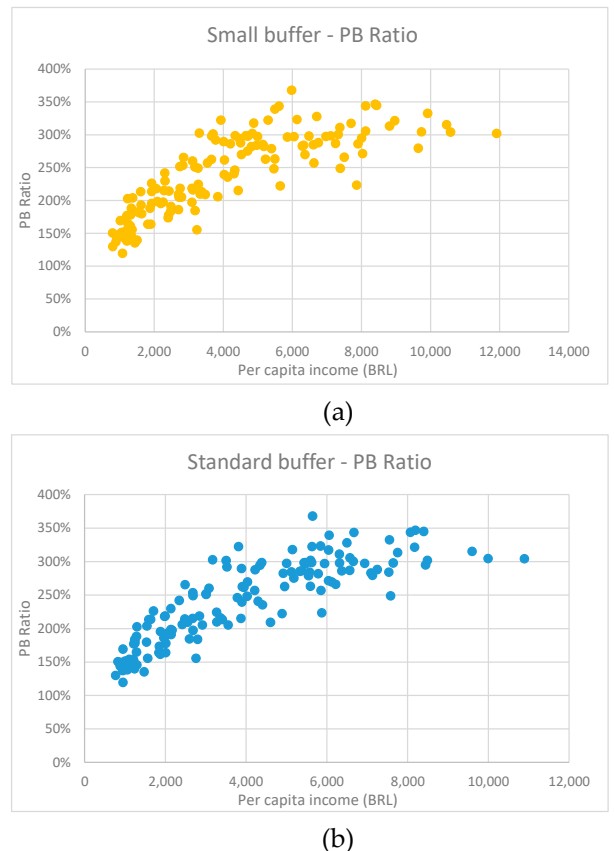

(a)

(b)

**Figure 4.** *Cont.*

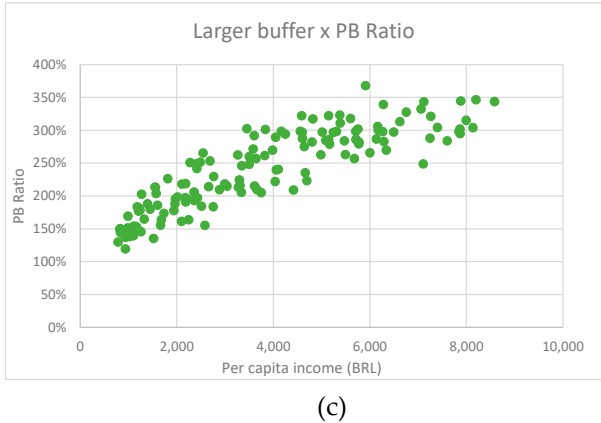

(c)

**Figure 4.** The plot of PB Ratio (as % of average sales, vertical axis) vs per capita income for each store influence buffer (in BRL per capita, horizontal axis): (**a**) small buffer; (**b**) standard buffer; (**c**) large buffer (see Table 1). Patterns are similar for all buffer sizes and indicate a positive correlation.

The plots in Figure 4 suggest a positive correlation. As average per capita income comes from a very detailed census data, it can be considered as the independent variable since its error should be lesser than the PB radius error.

Different functions (linear, semi-log, inverse, log) have a very good fit to that data and provide very low *p*-values for the slope parameter. However, only the log function ($Y = aX^b$, where Y is the PB ratio and X the per capita income; a and b are parameters) passes the Glejser heteroskedasticity test and its residue can be considered homoscedastic. Therefore, the log function can be statistically tested for positive correlation. Figure 5 shows residual plots for the results for standard buffer influence zone sizes for different regression models.

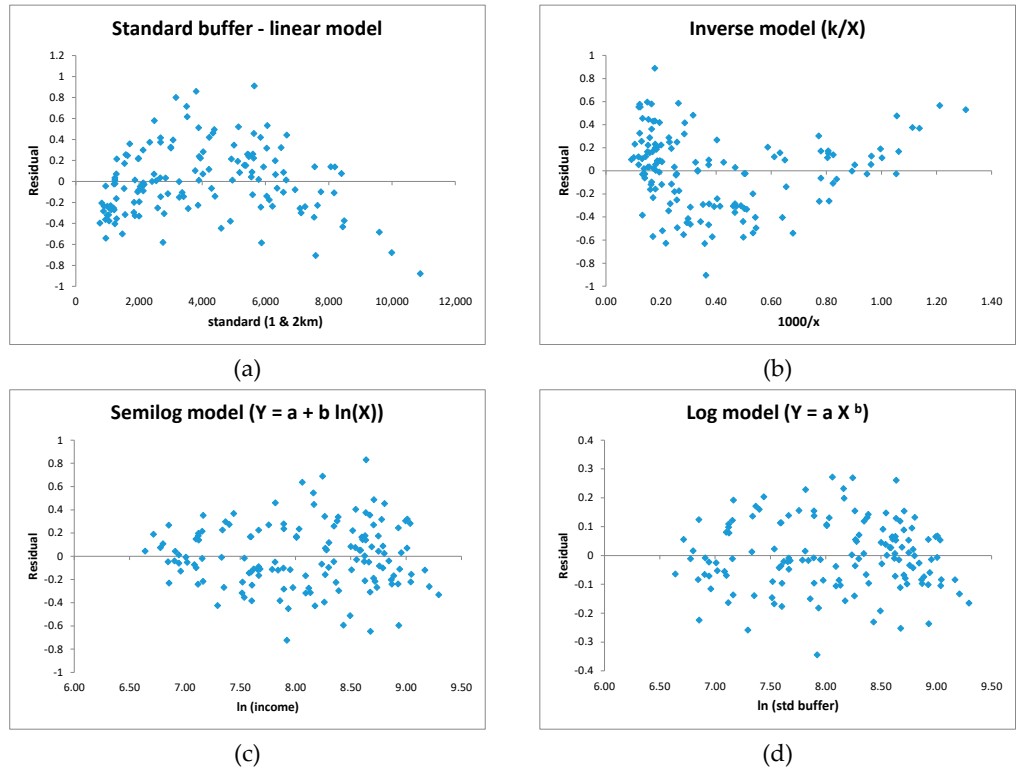

**Figure 5.** Residual plots for linear (**a**), inverse (**b**), semilog (**c**), and log (**d**) regression models; the three first plots (**a**,**b**,**c**) show evident residual correlation. Only the log model (**d**) passed the Glejser test.

The Michaelis–Menten ($Y = aX/(1 + bX)$) function also provides a very good fit for all buffer sizes (*p*-values < 2E-16), but as the function converges to a maximum value a/b, it would imply that there is a limit to the PB ratio (surprisingly in a very short range of 366% to 379% for all buffer sizes). The authors do not agree that there necessarily is an upper limit to the PB ratio in this particular income range and preferred to keep the log model.

The results for the log model are shown in Table 3. Figure 6 shows the regression curve for the standard buffer size.

**Table 3.** Results for the log ($Y = aX^b$) model, according to different buffer sizes. All show significant positive correlation.

| Influence Area Size | a | *p*-Value (a) | b | *p*-Value (b) |
|---|---|---|---|---|
| Small | $1.344 \times 10^{-1}$ | *** | $3.510 \times 10^{-1}$ | *** |
| Standard | $1.283 \times 10^{-1}$ | *** | $3.584 \times 10^{-1}$ | *** |
| Large | $1.147 \times 10^{-1}$ | *** | $3.735 \times 10^{-1}$ | *** |

Significance codes: 0 '***'. Source: Software R.

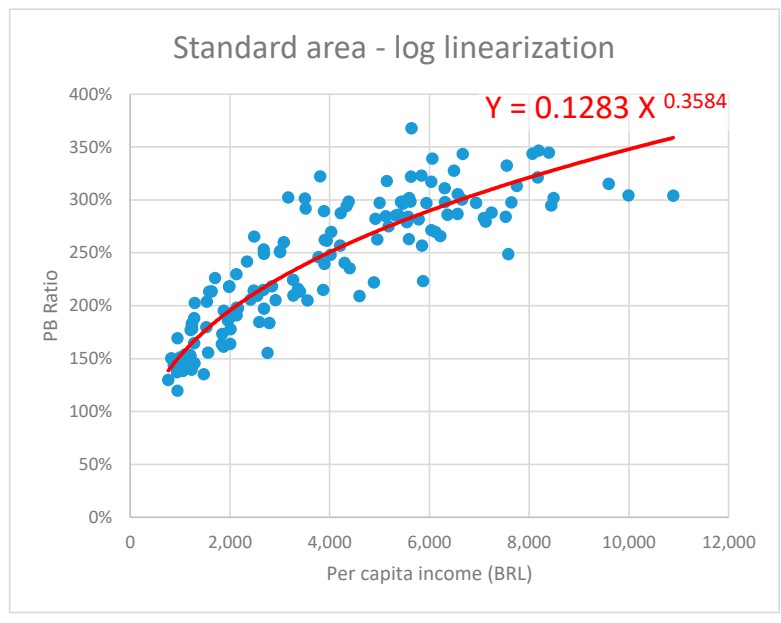

**Figure 6.** Log regression curve for PB ratio as a function of average per capita income of a store influence area.

Therefore, there is significant positive correlation between average income and panic buying.

## 5. Discussion and Conclusions

Based on sales data about toilet paper, the influence radius of stores, and the average per capita income in São Paulo city, this work concludes that there is a significant positive correlation between panic buying level and per capita income during the COVID-19 pandemic. Results are robust as sensitivity analysis performed using different buffer sizes generated the same conclusions. The log regression ($Y = aX^b$) showed the best fit to the data, with adequate residual patterns (homoscedastic), so making the statistical confidence intervals applicable.

Calculating the 95% confidence intervals for the regression (E(Y)) and the forecasting region (Y) at the lowest income zones (X = BRL$ 765, or 1.5 minimum wage; standard size buffer), the authors obtained intervals of [132%–145%] and [109%–175%], respectively. Thus, panic buying values are positive even for some of the lowest-income regions in São Paulo City. Hence, contrary to some beliefs in the media and academia [34,54,55], this research strongly indicates that this phenomenon is not a

behavior exclusive to more affluent people. This means that the stress of panic buying is pervasive in society and could be a particular source of anxiety to low-income people that do not have resources (cash, storage space) to fulfill this kind of demand.

Another interesting finding is the shape of the regression curve. The data plot suggests a concavity (the comet tail contour, see Figure 4) that regression analysis confirmed. Thus, panic buying does not grow linearly or faster with income but behaves like a strictly concave function with a diminishing slope. From a supply chain management perspective, this could allow for better demand forecast modeling at the establishment level. Additionally, as commented before, the authors chose log regression as they assume that there is no limit to panic buying within that income range, but an asymptotically converging function (as in Michaelis–Menten) could be considered if saturation is hypothesized.

Finally, the study presents some interesting figures. Panic buying could vary from as low as 140% to 360% daily sales for as long as ten days (see Figure 6 for standard size buffers) as a function of store influence area. There is a strong influence of income on panic buying indeed. It could be noted that Brazil did not have previous experiences of panic buying due to natural disasters, as in countries susceptible to hurricanes, typhoons, or earthquakes.

There are implications both to businesses and government decision-making during pandemics.

For retailers, this study provides information relevant to learning from the behavior of demand in times of crisis and to improve the management of inventory, replenishment, distribution centers, and storage space. They can also define better logistic strategies for dealing with panic buying products (such as rationing) to minimize stockouts and shortages on the shelves, as volume growth is properly sized as a function of influence areas.

Governments can also learn from this paper to consider the socioeconomic factors of panic and hoarding purchases, allowing the identification of places and neighborhoods where potential needs from the most vulnerable populations occur to avoid social panic, looting, and distress [58]. The need for protection mechanisms, such as donations (in-kind) or cash and voucher programs can be evaluated as well as rationing policies in cooperation with retailers. For instance, during the pandemics, a great deal of effort has been expended to deal with collection, storage, and distribution of food and hygiene items to low-income communities in São Paulo and elsewhere, with implications to humanitarian and disaster logistics management [59].

The research assumes that the average household income is representative of average behavior. Additionally, one limitation of the study is the focus on a single product. Toilet paper has been a symbol of panic buying during pandemics and general conclusions may apply to other typical disaster hoarding products, such as dry food and bottled water, but they may present different curves and thresholds. For instance, does panic buying level off at some specific income or simply slow down? In other words, is the Michaelis–Menten function fitter to model this phenomenon? Some hygiene and cleaning product categories have also experienced a surge in the same period, something that has seldom been seen before. Surely, the low-income population may better weight tradeoffs with toilet paper and prioritize other products, and it would shed more light on the assumption of the average behavior of lower-income classes. The authors intend to collect additional data to perform these multi-product analyses, including other large cities in Brazil.

**Author Contributions:** Conceptualization: H.T.Y.Y. and I.d.B.J.; methodology: H.T.Y.Y., I.d.B.J. and C.M.H.; software: C.M.H.; validation: H.T.Y.Y.; investigation: L.L.A. and M.C.R.P.; data curation: C.M.H., L.L.A. and M.C.R.P.; writing—original draft preparation: H.T.Y.Y. and I.d.B.J.; writing—review and editing: H.T.Y.Y. and I.d.B.J.; supervision: H.T.Y.Y. All the authors have read and agreed to the published version of the manuscript.

**Funding:** This research was funded by Coordination for the Improvement of Higher Education Personnel (CAPES), grant number Pró-Alertas 88887.091746/2014-01; and the National Council for Scientific and Technological Research (CNPq), grant number 313687/2019-6.

**Acknowledgments:** The authors would like to thank the grocery chains (names intentionally omitted), which provide us real transactions data, CAPES Coordination for the Improvement of Higher Education Personnel, and CNPq National Council for Scientific and Technological Research.

**Conflicts of Interest:** The authors declare no conflict of interest.

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
