# Peer review of "Relationship between Panic Buying and Per Capita Income during COVID-19"

_sustainability, doi:10.3390/su12239968_

Round 1

Reviewer 1 Report

The strengths of the article is original and interesting considerations with is consistent with the pattern of research. Solid methodology of the research with statistical analysis.

Therefore contribution to existing knowledge is considerable. Also advantage of the research is perfect organization & readability. I cannot find the weaknesses of the assessed article. Model article worthy of imitation.

In generally it is excellent article and very interesting considerations, which is consistent with the pattern of research. A very good article with the analysis of statistics on the topic under study.

Overall evaluation: article it is suitable for publication in current version.

Author Response

Thank you very much for your kind comments.

Point 1: Overall evaluation: article it is suitable for publication in current version.

Response 1: The authors appreciate your time and are intent to pursue this line of investigation with other basic, disaster-related, panic buying products.

Reviewer 2 Report

The paper under review appears as interesting, well-structured and it is (clearly) related to a current issue. A number of hypotheses have been put forward regarding hoarding during the confinement due to the COVID-19 pandemic and, specifically, toilet paper. However, most of such reasoning lacked a solid basis, and hence the relevance of having arguments and conclusions on a scientific basis (as it is the case in this paper). Thus, it could be considered for publication.

There are only a few comments / improvement suggestions prior to its definite acceptance:

  1. a) From a formal view, most bibliographical references are identified by a number in square brackets when quoted along the text (i.e. accordingly to the format rules for Sustainability). However, in some cases along pages 3 and 4 authors’ names and years are also included together to the number in square brackets and this should be avoided when aiming for a homogeneous writing.
  2. b) Most relevant, some of the main findings in the study is the positive correlation between average income per capita and panic buying, together to the fact that ‘panic buying happens in every income class, including low-income people’. This can be true (of course), but it implies assuming that ‘all’ people of ‘all’ income levels bought toilet paper in the different areas of the city (just in order to maintain the figure of ‘average income level’ for real customers). Whereas it could be that only people with higher (or lower) purchasing power bought this product in each one of considered areas. In other words, as we have not the information on the concrete income level per buyer, but only the average income level in each area, this would be an assumption which should be mentioned as a limitation of the study. To be precise, and for example, it is very likely that individuals with lower incomes hoarded more basic products, such as bottled water of other non-perishable foods. In this sense, the stated authors’ intention of collecting additional data to perform a multi-product analysis including other large cities in the country would be welcome. Even more, not only including large cities, but also small villages (or, even better, population centres of all / different sizes) would allow additional conclusions (for example, could São Paulo stores have attracted ‘additional’ buyers from smaller surrounding population centres to hoard concrete items?).

Author Response

Thank you very much for your comments and suggestions. They certainly contribute to improving the quality of the manuscript. Please see our replies below.

Point 1: From a formal view, most bibliographical references are identified by a number in square brackets when quoted along the text (i.e. accordingly to the format rules for Sustainability). However, in some cases along pages 3 and 4 authors’ names and years are also included together to the number in square brackets and this should be avoided when aiming for a homogeneous writing.

 Response 1: Thank you for your suggestion. We have corrected those points in the literature review along with pages 2, 3 and 4 and improved the readability.

Point 2: Most relevant, some of the main findings in the study is the positive correlation between average income per capita and panic buying, together to the fact that ‘panic buying happens in every income class, including low-income people’. This can be true (of course), but it implies assuming that ‘all’ people of ‘all’ income levels bought toilet paper in the different areas of the city (just in order to maintain the figure of ‘average income level’ for real customers). Whereas it could be that only people with higher (or lower) purchasing power bought this product in each one of considered areas. In other words, as we have not the information on the concrete income level per buyer, but only the average income level in each area, this would be an assumption which should be mentioned as a limitation of the study. To be precise, and for example, it is very likely that individuals with lower incomes hoarded more basic products, such as bottled water of other non-perishable foods. In this sense, the stated authors’ intention of collecting additional data to perform a multi-product analysis including other large cities in the country would be welcome. Even more, not only including large cities, but also small villages (or, even better, population centres of all / different sizes) would allow additional conclusions (for example, could São Paulo stores have attracted ‘additional’ buyers from smaller surrounding population centres to hoard concrete items?).

Response 2:  Thank you for your comment. We have added an observation regarding the implicit assumption in our discussion and conclusion chapter (lines 379, 387-388) and lessened the tone of our findings (lines 23-24, 335, 346-349), to make it explicit that we are talking about income classes and not individual people; it would be clear to the reader that we mean averages. Finally, the retailers agreed to furnish data on the other products, and in a few days, we would be able to work on that!

Reviewer 3 Report

This represents an interesting and relevant work on COVID-19 and cusumer behaviour. The only problem i have with this work is that there are several parallel publications coming up on COVID without proper account of what has been published on the rapidly emerging topic. For that matter i suggest the authors review previous contributions on this toipic and also set thier discussions in this emerging and braoder context. Here are some publicaitons on COVID-19 and consumption/sustainability etc.
Kanda, W., & Kivimaa, P. (2020). What opportunities could the COVID-19 outbreak offer for sustainability transitions research on electricity and mobility?. Energy Research & Social Science, 68, 101666.

Buechler, E., Powell, S., Sun, T., Zanocco, C., Astier, N., Bolorinos, J., ... & Rajagopal, R. (2020). Power and the Pandemic: Exploring Global Changes in Electricity Demand During COVID-19. arXiv preprint arXiv:2008.06988.

Sovacool, B. K., Del Rio, D. F., & Griffiths, S. (2020). Contextualizing the Covid-19 pandemic for a carbon-constrained world: Insights for sustainability transitions, energy justice, and research methodology. Energy Research & Social Science, 68, 101701.

Cheshmehzangi, A. (2020). COVID-19 and household energy implications: what are the main impacts on energy use?. Heliyon, 6(10), e05202.

Author Response

Thank you very much for your comments and suggestions. They certainly contribute to improving the quality of the manuscript. Please see our reply below.

Point 1: The only problem i have with this work is that there are several parallel publications coming up on COVID without proper account of what has been published on the rapidly emerging topic. For that matter i suggest the authors review previous contributions on this toipic and also set thier discussions in this emerging and braoder context. Here are some publicaitons on COVID-19 and consumption/sustainability etc.

Kanda, W., & Kivimaa, P. (2020). What opportunities could the COVID-19 outbreak offer for sustainability transitions research on electricity and mobility?. Energy Research & Social Science, 68, 101666.

Buechler, E., Powell, S., Sun, T., Zanocco, C., Astier, N., Bolorinos, J., ... & Rajagopal, R. (2020). Power and the Pandemic: Exploring Global Changes in Electricity Demand During COVID-19. arXiv preprint arXiv:2008.06988.

Sovacool, B. K., Del Rio, D. F., & Griffiths, S. (2020). Contextualizing the Covid-19 pandemic for a carbon-constrained world: Insights for sustainability transitions, energy justice, and research methodology. Energy Research & Social Science, 68, 101701.

Cheshmehzangi, A. (2020). COVID-19 and household energy implications: what are the main impacts on energy use?. Heliyon, 6(10), e05202.

Response 1: We have adjusted the literature review to address your points (lines 99 to 103) and incorporated your references, we appreciate your suggestion. Additionally, as we have already explored a lot of very recent literature related to COVID-19, they are now listed together. On the other hand, our study is focused on an old issue in humanitarian operations (is panic buying is the behavior of more affluent people?), and our main contribution is associated with that question. In a way, COVID is a very appropriate environment to research that, since we were fortunate enough to get data covering a large population which did not have storage issues at home, as the pandemic does not have the same infrastructure destruction effect as large disasters, such as earthquakes and hurricanes.
